# The Relationship between the Color of Electronic Protectors and the Outcome in Taekwondo Matches: Is There Fairness in National Competitions?

**DOI:** 10.3390/ijerph19127243

**Published:** 2022-06-13

**Authors:** Gennaro Apollaro, Pedro Vieira Sarmet Moreira, Yarisel Quiñones Rodríguez, Verónica Morales-Sánchez, Coral Falcó

**Affiliations:** 1Faculty of Medicine and Surgery, School of Sport Sciences and Exercise, University of Rome Tor Vergata, 00133 Rome, Italy; gen.2012.ita@hotmail.com; 2Biomedical Engineering Program, COPPE, Federal University of Rio de Janeiro, Rio de Janeiro 21941-592, Brazil; sarmet@peb.ufrj.br; 3Tech4Fight Sports Technology, Rio de Janeiro 21941-850, Brazil; 4Department of Social Psychology, Social Anthropology, Social Work and Social Services, University of Málaga, 29071 Malaga, Spain; yariselqr@gmail.com (Y.Q.R.); vomorales@uma.es (V.M.-S.); 5Department of Sport, Food and Natural Sciences, Western Norway University of Applied Sciences, 5020 Bergen, Norway

**Keywords:** color effect, red, electronic protector, technology, fair play, electronic scoring system

## Abstract

Although research on the effect of color in taekwondo has involved several international competitions, no previous study has investigated the presence of this phenomenon in national-level competitions. The main objective of this study was to analyze the relationship between the color protectors and success in 1155 taekwondo matches of the Italian and Uzbekistan Senior Championships (ITA-SC and UZB-SC) (2019 and 2021). The results showed no relationship between the color protectors and the match outcome, in both ITA-SC and UZB-SC (*p* = 0.71, *V* = 0.01; *p* = 0.61, *V* = 0.02). Moreover, no relationship emerged between the color protectors and the match outcome in the four editions of the SC. Stratifying analyses by weight category and sex, males showed positive relationships between the color blue/red and winning the match in 3 and 1 of 16 weight categories, respectively. Contrary, females showed positive relationships between the color blue/red and winning the match in 1 and 3 of 16 weight categories, respectively. Analyzing the two national contexts found that, in both the Italian and Uzbek contexts, matches in 2 and 2 of 16 weight categories were won by athletes wearing blue and red protectors, respectively. Significant relationships emerged between the color blue and winning the match with small asymmetry in the men’s UZB-SC and between the color red and winning the match with large asymmetry in the female ITA-SC. The implementation of the electronic point recording system for the body and head has had a positive impact on fairness in national taekwondo competitions, did not detect any effect of color related to cultural context, and did not allow for the color red to tip the scales between losing and winning in matches between athletes of similar ability and strength.

## 1. Introduction

The need to visually differentiate opponents is of utmost importance for players to separate “friend from foe”. For referees it is important to make the right calls, while for spectators, both in the arena as well as TV, it is to support their team [1]. In most Olympic combat sports (such as boxing, taekwondo, Greco-Roman wrestling, and freestyle wrestling), the colors of the uniforms and/or protectors used to differentiate the two athletes competing to win the match are blue and red [2]. In dual-contest sports, the importance of color for referees is not limited to making the right calls. Actually, color provides support in the crucial task of scoring attribution, helping referees to visually differentiate the short duration, high intensity, and often simultaneous techniques [3] performed by the two athletes. In this context, Hagemann et al. [4] hypothesized the presence of a psychological effect in referees triggered by the perception of colors that could lead to biases in the evaluation of identical performances. The hypothesis formulated by Hagemann et al. [4] arose primarily as a response to the pioneering theories on the effect of the color red in Olympic combat sports, formulated by Hill and Barton [5]. The latter authors showed that wearing red sportswear in boxing, taekwondo, and wrestling (Greco-Roman and freestyle) during the 2004 Athens Olympic Games (OOGG) had a positive and significant impact on the outcome of the match. The authors proposed that the advantage of wearing red was due to an evolutionary or cultural association with dominance and aggression [6,7,8]. When the researchers analyzed the various degrees of asymmetry, to quantify the role of confounding factors such as skill and strength between athletes, they found that there were significantly more red winners than blue winners only in matches between athletes of similar ability. That confirmed the key role of the color red when other factors (i.e., skill and strength) were fairly equal [5].

In contrast, the hypothesis of Hagemann et al. [4] attempts to highlight how the effect of red sportswear may have been wrongly attributed only to athletes and underestimated in the decision-making processes of referees. In their experiment, by having experienced referees judge the same taekwondo fights but with the colors of the protectors digitally reversed, Hagemann et al. [4] found that athletes in red received systematically more points than athletes in blue, even when their performance was identical. In response to the key role of the color red, when skill and strength are fairly equal [5], the authors argued that the referees’ decisions will be decisive when the athletes are relatively equal but will have relatively little influence when one is clearly superior to the other. Sorokowski et al. [9], by having Polish and Chinese students judge the same boxing fight in an experiment similar to Hagemann et al. [4], fully confirmed their results. They pointed out that the advantage of wearing red is not necessarily a consequence of the mechanisms linked to the two boxers fighting, but to the perception by observers as stronger, more aggressive, and more dominant than their opponents. Furthermore, considering the documented influence of culture on the perceptual meaning of colors [10,11,12], Sorokowski et al. [9] showed for the first time that the advantage of wearing red extends beyond Western sports culture. In fact, the authors hypothesized that colors might be perceived in different ways in different cultures in everyday conditions, but in a boxing match and other sporting competitions the emotional connotations might be similar, activating an evolutionary pattern in which the color red is related to aggressiveness and dominance.

To date, the debate on the effect of color [1,4,5,9,13,14] is still relevant in scientific research on Olympic combat sports as it relates to the fairness and fair play of the competition and consequently to the permanence of the sport on the Olympic circuit [15]. Although the suggestion to ban red sportswear was not taken up, the intuition of the technological aid to support referees [4] was almost inevitably welcomed. However, of the three Olympic combat sports originally investigated by Hill and Barton [5], taekwondo is the one that has undergone this process the most [16,17,18,19,20]. Taekwondo’s presence has been confirmed at the next OG in Paris 2024 and Los Angeles 2028 [21,22]. The renovations implemented over the previous and consecutive six participations of this sport in the world’s biggest sporting event provide more than one explanation for the current reconfirmations in the Olympic programmes. Firstly, we can identify these renovations as the rule changes, made by World Taekwondo (WT) to encourage more exciting and dynamic techniques, that have increased significantly since the first Olympic participation in Sydney 2000 [23]. Secondly, we can recognize even more of these renovations in the technological advances, put in place by the WT after the Beijing 2008 OG to make the sport fair and transparent, such as the implementation of an electronic point recording system for some (and then other) areas where scoring is allowed in taekwondo [23].

The first technological innovation, the electronic body protector, made its entrance into the Olympic scene at London 2012 [24]. Carazo-Vargas and Moncada-Jiménez [16] and Falcó et al. [18], analyzing the relationship between the color of the protectors and the success in the matches in the 2013 World Championships (WC) and the 2012 OOGG (including qualifying tournaments), respectively, when electronic body protectors were used, reported overall greater objectification and fairness in competitions in line with previous hypotheses [4,25,26] and underlined the need to continue with the implementation of a fully objective system. Subsequently, an electronic head protector was introduced in the Olympic competition in Rio 2016 [27]. Therefore, to date, in taekwondo the manual scoring system by referees is limited to punch techniques and penalties (but with an Instant Video Replay system to support athletes and coaches; [23]). Apollaro and Falcó [20], studying the relationship between the color of the protectors and the success in the matches of six World Grand Prix Series (WGPS) over two Olympic 4-year periods (2015 and 2018) when electronic head protectors were also used, confirmed and extended the findings of the two previous studies [16,18], and overall outlined a current framework “deliberately” different from that of the Athens 2004 OOGG by Hill and Barton [5].

Although research on the effect of color in taekwondo has involved several international competitions (e.g., OOGG, WC, WGPS), to our knowledge no previous study has investigated the presence of this phenomenon in national-level competitions. Firstly, considering that electronic body and head protectors have been introduced in national competitions in parallel with international competitions, a study investigating the impact of technological advances on the fairness of national competitions could be important as victory in these competitions generally represents the entry step into the high level for taekwondo athletes. Secondly, a study exploring the effect of color in sport in specific (rather than in international competitions with athletes and referees from all over the world) and different (not only in western culture) cultural contexts would add information to this area of research at its early stage [9]. Therefore, the main objectives of this study are: (1) to analyze the relationship between the color of the protectors and the success in the matches of the Italian Senior Championships (ITA-SC) and the Uzbekistan Senior Championships (UZB-SC) (2019 and 2021); and (2) to analyze the effect of confounding factors, such as skill and strength, among the athletes through the analysis of different degrees of asymmetry. In line with the recent results of Apollaro and Falcó [20], we hypothesize that the number of victories in national competitions by the athletes in blue should be similar (or equal) to the number of victories achieved by the athletes in red, as electronic body and head protectors was also used in these competitions. For the same reason, we do not hypothesize any effect of color related to cultural context but, in case it is present, we hypothesize a similar direction of the effect in the two cultural contexts, in line with Sorokowski et al. [9]. Lastly, no relationship should emerge between the color of the electronic protectors and the success in the matches between athletes with similar skill and strength, in contrast to the findings using the manual scoring system [4,5].

## 2. Materials and Methods

### 2.1. Methodology and Study Design

This study was conducted using observational methodology with a mixed-methods research design. The color of the electronic protectors (blue and red) was considered as the independent variable. The dependent variable was the outcome of the match (win and defeat).

### 2.2. Participants

The present study included a total of 1155 matches, corresponding to two ITA-SC and two UZB-SC: the 2019 ITA-SC (held in Casoria, 14–15 December 2019; *N* = 265), the 2021 ITA-SC (held in Busto Arsizio, 20–21 November 2021; *N* = 273), the 2019 UZB-SC (held in Gulistan, 24–25 December 2019; *N* = 342) and the 2021 UZB-SC (held in Tashkent, 4–6 November 2021; *N* = 275). Data were collected from publicly available online sources (https://www.tpss.eu/login.asp (accessed on 28 November 2021) and https://www.ma-regonline.com/ (accessed on 15 December 2021)), ensuring the anonymity and privacy of individual athletes. The use of data from open access sites has been previously described in other studies [18,20] and there are no ethical issues involved in the analysis and interpretation of the data used as these were obtained in a secondary form and not from direct experimentation.

### 2.3. Measures

In the analysis, the color (blue or red) of the protectors (Dae do^®^, L’Hospitalet de Llobregat, Spain: 2019 ITA-SC and 2021 ITA-SC; KPNP^®^, Seoul, Korea: 2019 UZB-SC and 2021 UZB-SC), the national context (Italy or Uzbekistan), the sex of the athlete (male or female), and the weight category (fin, fly, bantam, feather, light, welter, middle, and heavy) were included. Subsequently, the different degrees of relative ability (asymmetry) were also taken into consideration. The details of the sample are given in Table 1.

### 2.4. Statistical Analysis

Data were tabulated and organized in a Microsoft Excel worksheet and then reported and analyzed using IBM SPSS Statistics for Windows, version 26.0 (IBM Co., Armonk, NY, USA). The effect of wearing blue or red protectors was examined according to national context, sex, weight categories, and asymmetry classes. To this end, chi-square (χ^2^) tests of association were performed to identify the associations between the color of the protectors and the winning of the match. The strength of the association was assessed using Pearson’s contingency coefficient (*C*) and the threshold values were classified as weak (0.0–0.3), moderate (0.3–0.6), and strong (>0.6) [28]. Effect size (ES) was reported using Cramer’s *V* as weak (0.05–0.09), moderate (0.10–0.14), strong (0.15–0.24), and very strong (>0.25) [29]. When the initial χ^2^ test of association was found to be statistically significant, odds ratios (OR; calculated through Equation (1)) and 95% confidence intervals (95% CI) were calculated [30]. In line with Hill and Barton [5], matches were recorded into different classes of asymmetry on the basis of the difference in points scored by each athlete. Every National Senior Championships was categorized on the basis of the quartile of the final points difference in the match, where the first quartile of points difference represents symmetrical contests between athletes of similar ability and the fourth quartile represents contests between athletes with large asymmetries in ability. Matches stopped early (“Referee Stops Contest”) were scored as highly asymmetric contests and coded in the fourth quartile. χ^2^ goodness-of-fit tests were performed to compare the observed distribution of the different asymmetry classes with the expected distribution. Also, for these analyses, ES was reported using Cramer’s *V*. Statistical significance was accepted at *p* < 0.05.
OR = (nBW)^2^/(nRW)^2^,(1)
where nBW is the number of Blue Winners and the nRW is the number of Red Winners.

## 3. Results

For the whole samples (Table 2), the results showed non-significant relationships between the color of the protectors and the outcome of the match, in both ITA-SC and UZB-SC.

At the same time, for the individual editions of the National Senior Taekwondo Championships (Table 2), the results showed no relationship between the color of the protectors and the outcome of the match in the 2019 ITA-SC, 2021 ITA-SC, 2019 UZB-SC, and 2021 UZB-SC.

Based on the absence of significant associations in the total samples and in the individual editions, the subsequent analyses of sex and weight categories were carried out according to the national context.

The results showed that for males (Table 3), there was a very strong relationship between individuals wearing blue protectors and winning the match in the heavyweight category during the ITA-SC (*C* = 0.37; OR = 5.44, 95% CI = 1.41–21.05), strong relationships in the featherweight category during the ITA-SC (*C* = 0.19; OR = 2.19, 95% CI = 1.07–4.49) and in the fin weight category during the UZB-SC (*C* = 0.23; OR = 2.66, 95% CI = 1.35–5.23). In contrast, the results showed a very strong relationship between wearing red protectors and winning the match in the fin weight category during the ITA-SC (*C* = 0.27; OR = 0.32, 95% CI = 0.12–0.84).

For females (Table 3), the results showed a very strong relationship between individuals wearing blue protectors and winning the match in the lightweight category during the UZB-SC (*C* = 0.32; OR = 4.00, 95% CI = 1.37–11.70). On the contrary, very strong relationships were found between wearing red protectors and winning the match in the flyweight category during the ITA-SC (*C* = 0.38; OR = 0.17, 95% CI = 0.06–0.49), in the bantamweight category during the UZB-SC (*C* = 0.32; OR = 0.25, 95% CI = 0.09–0.70) and in the welterweight category during the UZB-SC (*C* = 0.37; OR = 0.18, 95% CI = 0.05–0.71).

Subsequently, the observed distribution of the different classes of skewness was compared with the expected distribution. In addition, the relationship between the color of the protectors and the winning of the match was analyzed for each degree of asymmetry (none, small, medium, and large) according to national context and sex (Figure 1).

In the male ITA-SC, the different degrees of asymmetry were equally distributed, but no significant relationship was found between the color of the protectors and the outcome of the match with none, small, medium, and large asymmetry. In the female ITA-SC, the different degrees of asymmetry were equally distributed, and the results showed a very strong relationship between wearing red protectors and winning the match with large asymmetry (*C* = 0.30; OR = 0.27, 95% CI = 0.11–0.70). In the male UZB-SC, the different degrees of asymmetry were equally distributed, and the results showed a moderate relationship between wearing blue protectors and winning the match with small asymmetry (*C* = 0.14; OR = 1.76, 95% CI = 1.00–3.08). Lastly, in the female UZB-SC, the different degrees of asymmetry were equally distributed, but no significant relationship was found between the color of the protectors and the outcome of the match with none, small, medium, and large asymmetry.

## 4. Discussion

When analyzing the total samples (2019/2021 Italian Senior Championships and 2019/2021 Uzbekistan Senior Championships), our results are in line with Apollaro and Falcó [20] who investigated WGPS, held in a timeframe immediately preceding ours (2015 and 2018), when electronic body and head protectors were also used. Apollaro and Falcó [20] included 1327 matches and their results showed a non-significant relationship between the color of the protectors and the outcome of the match. Previously, Carazo-Vargas and Moncada-Jiménez [16] and Falcó et al. [18], analyzing 718 matches from the 2013 WC and 462 matches from the 2012 OOGG (including the 2011 and 2012 Olympic qualifying tournaments) when only body electronic protectors were used, found similar percentages between winners with blue and red protectors, and a non-significant relationship between the color of the protectors and the outcome of the match, respectively, were found. Taken together these results, it seems to outline, from national to top level, a competitive picture characterized by fairness and objectivity over the last decade. The hypothesis several times formulated [16,18,20], regarding the role of an electronic point recording system for areas where scoring is allowed [23], justifies these results, that are also confirmed in our study. In support of this, technological innovations began after the 2008 OOGG [23] finding in fact in the 2012 OOGG the first supporting evidence [18]. In contrast, a positive and significant impact of the color red on the outcome of the match in taekwondo was previously found at the 2004 OOGG, when a scoring system totally dependent on the referees was used [5]. Furthermore, the advantage of wearing red has been systematically reported in studies that simulated scoring with the manual system [4,9].

Stratifying the analysis by individual competitions, our results showed no relationship between the color of the protectors and the outcome of the match in the four editions of the National Championships. Apollaro and Falcó [20] in their study found a positive relationship between wearing blue protectors and winning the match in 2018 WGPS-1 and between wearing red protectors and winning the match in 2015 WGPS-2, but no relationship in the other four editions considered. Carazo-Vargas and Moncada-Jiménez [16] found a similar percentage among winners with blue and red protectors in the only edition of the WC (2013) included in their study. In contrast, Falcó et al. [18] found a positive relationship between wearing red protectors and winning the match in the 2011 Asian and 2012 European Qualification tournaments, but no relationship in the other two competitions analyzed. When analyzed together, the effect of the color red is present in 1 of 10 competitions where electronic head protectors were also used (the present study and Apollaro and Falcó, [20]), and in two of five competitions when only electronic body protectors were used [16,18]. In this context, the analysis by individual competitions shows the gradual decrease of the effect of the color red over the last decade, highlighting its minimized presence with the use of the electronic point recording system, also for the head. This is in line with the fact that, to date, the manual scoring system by referees is limited to punch techniques and penalties (but with an Instant Video Replay system to support athletes and coaches; [23]). Therefore, the analysis for individual competitions seems to present a greater sensitivity in detecting the further positive impact of the introduction of the electronic helmet on the fairness of taekwondo competition. On the contrary, the previous analysis of the total samples (when taken individually) could erroneously lead to the assumption of the total disappearance of the phenomenon in the last decade with a clear change starting from the 2012 OOGG. In line with the conclusions of Apollaro and Falcó [20], our study confirms that the further introduction of the electronic helmet has made the scoring system more objective with very similar winning percentages (between blue and red) when analyzing the total sample and an even more balanced situation (between blue and red) when analyzing individual editions of the Senior National Championships.

Regarding the analysis of the weight categories, Apollaro and Falcó [20] showed a slight shift tendency for the color effect, by switching from red to blue, when comparing their results with previous results by Falcó et al. [18]. They [20] hypothesized that when the referees’ influence in judging performance is reduced to punches and penalties, and following the color-in-context theory [31], seeing red has a stronger influence to increase performance than wearing red. Firstly, in our study the number of classes with a significantly higher percentage of red winners further confirms in the weight categories the gradual decrease over time of the referees’ advantage towards those wearing this color. Secondly, the inter-class equity situation in favor of blue and red found in our results does not confirm the slight tendency for the color effect to shift from red to blue [20]. The current data, when combined with more recent results in which the same system was used [20], seem to indicate that some weight categories are generally more sensitive to seeing the color red, while others are more influenced by wearing it. Our hypothesis is that when the influence of referees in judging performance is minimized, wearing blue (and seeing red) has more influence than in the past [5,18], but similar to the current influence of wearing red in increasing performance.

Stratifying the analysis of the weight categories for each sex, Apollaro and Falcó [20] highlighted that males were slightly more sensitive or seemed to be more influenced by the color of the protectors than females. Falcó et al. [18] found a very similar, but completely opposite situation, where females seemed to be more influenced by color than males. The results of our study seem to lie in the middle in that males and females are affected by color in the same proportion. In this context, if on the one hand it is not possible to determine which sex is more sensitive to the effect of color, on the other hand the effects of blue and red would seem to follow two distinct and opposite directions between sexes. The current data, when combined with the results of previous studies [18,20], show a mirror-image situation in which the blue and red color effect is present in 11 and 4 male and 4 and 11 female weight categories, respectively. Moreover, this situation persists even when only the most recent competitions are taken into account (in which electronic head protectors were also used; the present study and Apollaro and Falcó [20]) with the effect of the color blue and red present in 9 and 3 male and 3 and 6 female weight categories, respectively. From this framework, although there is evidence of a gradual decrease over time of the referees’ advantage over the wearer of red (in both males and females), it would appear that wearing blue (and seeing red) has an influence in increasing performance in males, and wearing red benefits females. These results complete and extend what was found and hypothesized in the previous weight categories analysis, confirming the importance of stratifying the analysis of the effect of color by weight categories for each sex. Indeed, while generally some weight categories seem to be more sensitive to noticing color and others more influenced by wearing color, sex would seem to add information about the direction of these two phenomena.

To our knowledge, the only attempt to investigate the effect of color in combat sports outside, in part, Western culture goes back to Sorokowski et al. [9]. In their experiment, they had Polish and Chinese students judge the same boxing match (in which the performance level of both boxers was very similar) using a manual scoring system, but with the colors digitally reversed. Firstly, they confirmed that generally the athlete in red received more points than the athlete in blue, as a consequence of a mechanism related to the observer’s perception [4]. Secondly, they demonstrated for the first time that the advantage of wearing red extended beyond western sporting culture and was equally strong in Poland and China as a potential cultural universal. In our study, the situation of equity between weight categories in favor of blue and red found equally in the two cultural contexts, when combined with the absence of relationships between the color of the protectors and the outcome of the match in the total samples and the individual editions, confirms our initial hypothesis. That is, there is no color effect related to cultural context as the use of electronic protectors (for the body and head) made both national competitions fair and transparent, in line with what was found in international competitions that used the same system [20]. However, further investigating the effect of color by sex in the two national contexts would seem to repeat the pattern identified in the previous analysis. Italian and Uzbek males show the blue and red color effect in 2 and 1 and 1 and 0 classes, respectively. In a mirror-image situation, Italian and Uzbek females show the blue and red color effect in 0 and 1 and 1 and 2 classes, respectively. These results further support our hypothesis that the current reduced influence of wearing red in increasing performance, in conjunction with the similar influence of wearing blue (and seeing red), may be more a consequence of the mechanisms associated with the two athletes fighting and that sex justifies the direction of the two phenomena.

Hill and Barton [5], analyzing the different degrees of asymmetry to quantify the role of confounding factors such as skill and strength between the athletes, included men’s boxing, taekwondo, and wrestling (Greco–Roman and freestyle) matches from the 2004 OOGG in the analysis. They found that there were significantly more red winners than blue winners only in matches between athletes of similar ability, with the red advantage that seemed to decrease as asymmetries in competitive ability increased. Hill and Barton [5] confirmed the key role of the color red when the other factors were fairly equal. In response to these results, Hagemann et al. [4], by having experienced referees judge the same taekwondo matches but with the colors of the protectors digitally reversed, found that athletes in red systematically received more points than athletes in blue, even when their performances were identical. In response to the key role of the color red when skill and strength are fairly equal [5], the authors argued that referees’ decisions will be decisive when athletes are relatively equal but will have relatively little influence when one is clearly superior to the other. To our knowledge, the present study was the first to investigate the different degrees of asymmetry in taekwondo competitions using the electronic system and to extend the analyses to the female sex as well. Although Carazo-Vargas and Moncada-Jiménez [16] took asymmetry into account, they did not specifically analyze it but included it as one of the variables in the regression analysis. In contrast, Falcó et al. [18] and Apollaro and Falcó [20] quantified the role of confounding factors by studying the influence of being a seeded athlete. Considering the limitations of this method when analyzing competitions in which the ranking of the athletes is similar (e.g., WGPS) and that the ranking is based on points acquired in the previous 4-year Olympic period, Apollaro and Falcó [20] proposed the method of Hill and Barton [5] as a valid alternative since it is based on the degree of skill and strength of the athlete on the day of a given competition. Furthermore, in national competitions the asymmetry method is appropriate as these competitions do not use the ranking. Our results confirm the initial hypothesis: there is no relationship between the color of the electronic protectors and the success in matches between athletes of similar skill and strength, i.e., red does not tip the scales between losing and winning in these matches compared to when a scoring system totally up to the referees was used [5]. This corroborates the hypotheses of Hagemann et al. [4] and highlights once again how in the past the effect of red has been wrongly attributed only to athletes and underestimated in the decision-making processes of referees.

### 4.1. Limitations and Future Research Lines

Although this study was the first to investigate the effect of color in national-level taekwondo competitions, some limitations and lines of future research should be considered. Firstly, we only included two national contexts in the analyses as the results of national competitions, which are retrievable from publicly available online sources and are limited compared to international competitions. The possibility in the future to involve more national contexts in the analyses would allow for an extension of the cross-cultural comparison in combat sports, adding information to this area of research at its early stage [9]. Secondly, archival research should be continuously updated to assess whether previous results stand the test of time [1]. Therefore, future studies could include in the analysis new editions of the national championships we analyzed and/or further competitions in which electronic protectors (for the body and head) were used to confirm the results. Finally, Sorokowski et al. [9] pointed out that the great majority of studies on the effect of color in sport are simply correlations or observations. Goldschmied et al. [1] also highlighted the problem of using sports performance or even the outcome of the competition as a dependent variable in color effect studies, as sports performance is influenced by a wide range of interacting factors. To this end, future research on the effect of color in combat sports should focus on conducting experimental studies (1) on observers [4,9,32]; (2) on the relationship between color and motor/sports performance (and physiological parameters) of athletes such as gross and fine motor skills [33], strength production [34,35], heart rate [36], and cortisol and oxygen consumption [37].

### 4.2. Practical Applications

Until the specific mechanisms behind the benefits of wearing and/or seeing a certain color are known [1], the suggestion to ban a color from sports competitions will probably not be accepted. Therefore, the technological aid to support the referees is currently a fundamental tool and its continuous evolution might also enable the objectification of punch techniques in the future. On the other hand, we support the recommendations of Falcó et al. [18], to Taekwondo coaches and athletes, to pursue psychological training that facilitates the athlete’s self-confidence (making them independent of the color of the protectors) and, considering the results of the present study, that is specific to males and females.

## 5. Conclusions

The implementation of the electronic point recording system for the body and head has also had a positive impact on fairness in national taekwondo competitions, confirming it as a good tool for decreasing the referees’ advantage towards those wearing red. The current data, combined with more recent data in which the same system was used, seem to outline a balanced situation in the weight categories where wearing blue (and seeing red) has more influence than in the past but is similar to the current reduced influence of wearing red in increasing performance. Considering that the influence of referees is minimized, these phenomena may be more a consequence of the mechanisms associated with the two athletes fighting, with sex seeming to add information about the direction of the two phenomena. Finally, objectification of the scoring system did not detect any effect of color related to cultural context and did not allow for the color red to tip the scales between losing and winning in matches between athletes of similar ability and strength.

## Figures and Tables

**Figure 1 ijerph-19-07243-f001:**
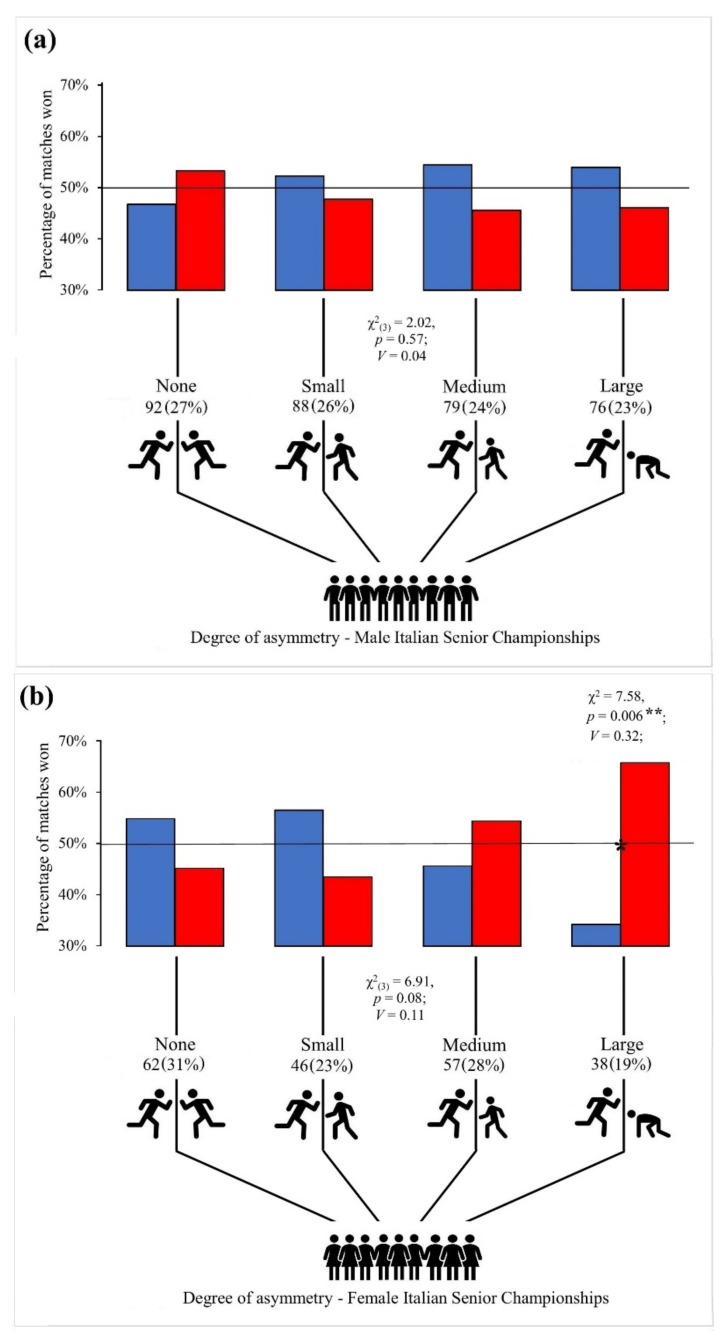
Percentage (%) of taekwondo matches won by athletes according to their color of protectors (blue and red) given different degrees of relative ability (asymmetry) in the two athletes in each match. (**a**) Male ITA-SC; (**b**) Female ITA-SC; (**c**) Male UZB-SC; (**d**) Female UZB-SC. Black lines at 50% indicate the expected percentage of wins by blue and red under the null hypothesis that the color is not associated with the outcome of the match. χ^2^: chi-square; * significant association between the color of the protectors and the winning of the match (* *p* < 0.05; ** *p* < 0.01); *V*: Cramer’s *V*.

**Table 1 ijerph-19-07243-t001:** Number of matches in each National Senior Taekwondo Championships.

Edition	Sex	Weight Category							
		Fin −54 kg M −46 kg F (*N* = 167)	Fly 54.1–58 kg M 46.1–49 kg F (*N* = 190)	Bantam 58.1–63 kg M 49.1–53 kg F (*N* = 220)	Feather 63.1–68 kg M 53.1–57 kg F (*N* = 182)	Light 68.1–74 kg M 57.1–62 kg F (*N* = 174)	Welter 74.1–80 kg M 62.1–67 kg F (*N* = 91)	Middle 80.1–87 kg M 67.1–73 kg F (*N* = 66)	Heavy +87.1 kg M +73.1 kg F (*N* = 65)
2019 Italian Senior Championships (*N* = 265)	M (*N* = 166)	18	23	41	26	26	11	9	12
	F (*N* = 99)	9	15	25	17	13	8	6	6
2021 Italian Senior Championships (*N* = 273)	M (*N* = 169)	18	31	31	36	30	6	9	8
	F (*N* = 104)	13	19	23	17	15	8	5	4
2019 Uzbekistan Senior Championships (*N* = 342)	M (*N* = 232)	43	34	40	33	34	20	18	10
	F (*N* = 110)	18	18	18	12	19	12	6	7
2021 Uzbekistan Senior Championships (*N* = 275)	M (*N* = 173)	28	29	27	25	26	18	7	13
	F (*N* = 102)	20	21	15	16	11	8	6	5

M: Male; F: Female.

**Table 2 ijerph-19-07243-t002:** Percentage of winners wearing blue and red protectors in each National Senior Taekwondo Championships.

Edition (Number of Matches)	Color Protectors		Chi-Square, *p*; Pearson’s *C*; Cramer’s *V*
	Blue (%)	Red (%)	
2019 Italian Senior Championships (*N* = 265)	53.6	46.4	χ^2^ = 2.73, *p* = 0.10; *C* = 0.07; *V* = 0.07
2021 Italian Senior Championships (*N* = 273)	47.6	52.4	χ^2^ = 1.24, *p* = 0.27; *C* = 0.05; *V* = 0.05
Overall (*N* = 538)	50.6	49.4	χ^2^ = 0.13, *p* = 0.71; *C* = 0.01; *V* = 0.01
2019 Uzbekistan Senior Championships (*N* = 342)	48.8	51.2	χ^2^ = 0.37, *p* = 0.54; *C* = 0.02; *V* = 0.02
2021 Uzbekistan Senior Championships (*N* = 275)	49.8	50.2	χ^2^ = 0.007, *p* = 0.93; *C* = 0.004; *V* = 0.004
Overall (*N* = 617)	49.3	50.7	χ^2^ = 0.26, *p* = 0.61; *C* = 0.02; *V* = 0.02

**Table 3 ijerph-19-07243-t003:** Percentage of taekwondo matches won by male and female athletes according to their color of protectors, national context, and weight categories.

Weight Category	2019/2021 Italian Senior Championships (*N* = 538)		2019/2021 Uzbekistan Senior Championships (*N* = 617)	
	Male (%) (*N* = 335) Chi-square, *p*; Cramer’s *V*	Female (%) (*N* = 203) Chi-square, *p*; Cramer’s *V*	Male (%) (*N* = 405) Chi-square, *p*; Cramer’s *V*	Female (%) (*N* = 212) Chi-square, *p*; Cramer’s *V*
Fin	Red: 63.9 (36) χ^2^ = 5.56, *p* = 0.018 *; *V* = 0.28	Red: 59.1 (22)	Blue: 62 (71) χ^2^ = 8.14, *p* = 0.004 **; *V* = 0.24	Blue/Red 50 (38)
Fly	Blue: 57.4 (54)	Red: 70.6 (34) χ^2^ = 11.53, *p* = 0.001 **; *V* = 0.41	Red: 54 (63)	Red: 53.8 (39)
Bantam	Red: 55.6 (72)	Blue: 54.2 (48)	Red: 55.2 (67)	Red: 66.7 (33) χ^2^ = 7.33, *p* = 0.007 **; *V* = 0.33
Feather	Blue: 59.7 (62) χ^2^ = 4.65, *p* = 0.031 *; *V* = 0.19	Blue/Red: 50 (34)	Blue: 53.5 (58)	Blue: 53.6 (28)
Light	Blue/Red: 50 (56)	Blue: 60.7 (28)	Red: 58.3 (60)	Blue: 66.7 (30) χ^2^ = 6.67, *p* = 0.01 **; *V* = 0.33
Welter	Red: 52.9 (17)	Blue: 62.5 (16)	Blue: 52.6 (38)	Red: 70 (20) χ^2^ = 6.40, *p* = 0.01 **; *V* = 0.40
Middle	Blue: 55.6 (18)	Blue: 54.6 (11)	Blue: 56 (25)	Blue/Red 50 (12)
Heavy	Blue: 70 (20) χ^2^ = 6.40, *p* = 0.011 **; *V* = 0.40	Red: 60 (10)	Blue: 52.2 (23)	Red: 66.7 (12)

(*N*): the number of matches in each weight category; * *p* < 0.05, ** *p* < 0.01: significant association between the color of the protectors and the winning of the match.

## Data Availability

Not applicable.

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
