# Peer review of "The Relationship between the Color of Electronic Protectors and the Outcome in Taekwondo Matches: Is There Fairness in National Competitions?"

_ijerph, 2022, doi:10.3390/ijerph19127243_

Round 1

Reviewer 1 Report

The main objective of this study was to analyze the relationship between the color protectors and success in 1,155 taekwondo matches of the Italian and Uzbekistan Senior Championships (2019 and 2021). The results showed no relationship between the color protectors and the match outcome, in both ITA-SC and UZB-SC (p = 0.71, V = 0.01; p = 0.61, V = 0.02). Moreover, no relationship emerged between the color protectors and the match outcome in the four editions of the SC. Males showed positive relationships between the color blue/red and winning the match in 3 and 1 of 16 weight categories, respectively. Contrary, females showed positive relationships between the color blue/red and winning the match in 1 and 3 of 16 weight categories, respectively. Both in the Italian and Uzbek context, matches in 2 and 2 of 16 weight categories were won by athletes wearing blue and red protectors, respectively. Significant relationships emerged between the color blue and winning the match with small asymmetry in the men's UZB-SC and between the color red and winning the match with large asymmetry in the female ITA-SC. The implementation of the electronic point recording system has had a positive impact on fairness in national taekwondo competitions, did not detect any effect of color related to a cultural context, and did not allow for the color red to tip the scales between losing and winning in matches between athletes of similar ability and strength.

Title: the title properly explains the purpose and objective of the article

Abstract: abstract contains an appropriate summary for the article, the language used in the abstract is easy to read and understand, and there are no suggestions for improvement.

Introduction: authors do provide adequate background on the topic and reason for this article and describe what the authors hoped to achieve.

Results: the results are presented clearly, the authors provide accurate research results, and there is sufficient evidence for each result.

Conclusion: in general: Good and the research provides sample data for the authors to make their conclusion.

Grammar: Need Some revision. (Check The Paper Comments).

Please provide the following information :  

what is the novelty of this paper compared to previous studies?

Reviewer 2 Report

Thank you for the opportunity to review an interesting topic „The impact of the electronic scoring system on outcome fairness in taekwondo matches: the case of two national contexts“. The main objective of this study was to analyse the relationship between the color protectors and success in 1,155 taekwondo matches of the Italian and Uzbekistan Senior Championships (2019 and 2021).

Although the results of the study are interesting, I have major and minor concerns:

Major concerns

L 134: It is necessary to fix the Methods Unit and specify what is the design of the study? The Authors must to indicate what survey (quantitative or qualitative) has been carried out? Also, what type of study was described in the manuscript (in terms of experiment or observation)? If the type of study is monitored, it is necessary to specify whether this was a cross-sectional study? Additionally, it is necessary to calssify and describe both dependent and independent variables.

L 158: Chi-square (χ2) tests do not indicate the association. This test is designed to identify the differences between the variables have been analysed. I don't recommend writing chi-square values in the Results Unit as only the correlation coefficient is sufficient.

L 203-209: There remains uncertainty as to the purpose for which the triple statistical indicators are written in the text, for example, „χ2 = 7.33, p = 0.007; C = 0.32; V = 0.33; OR = 0.25, 95% CI = 0.09–0.70“. To my knowledge, only the correlation coefficient is used to indicate the relationship between the variables.

Tables 2 and 3 must be supplemented by a clear record of the statistical indicators and the statistical reliability in numerals (for example, “p = 0.005”, etc.).

Figure 1 must be corrected without writing all the statistical indicators. The title of the figure must be abbreviated. All of this information must be written in the Results Unit.

L 247-253: It is not recommended to overwrite the study aims in the Discussion Unit.

L 329: Did the Authors make “gender analysis”? It seems necessary to revisit the Discussion Unit and check it for the errors of logical wheel. Furthermore, it is necessary to systematise and discuss the substantive results of this study. Information needs to be systematized and shortened. You don't have to rewrite your own results or those of other Authors.

L 431: The Authors have not identified the effect. The design of this study (non-experimental study) can only suggest the correlation.

L 453: The Authors did not identify “impact,” “influence.” The authors have only identified the correlation between variables. It seems necessary to rewrite the Conclusions. The main results of this study must be generalized and systematically updated in the Conclusions.

Minor concerns

It seems necessary to update a title of the paper. The authors did not conduct a case study or analyze the impact. There is a requirement to write sentences of background in the Abstract. What does the keyword “cross-cultural” mean?

L 37: What type of wrestling do the Authors write about in the document? It seems necessary to clarify whether is it freestyle wrestling or Greco-Roman wrestling.

L 49: The Authors must write a reference at the end of the sentence.

L 150, 153: There is a strong recommendation to specify the weight categories in kilograms in Table 1 (Fly „X – Y kg”, Bantam „X – Y kg” and etc.).

L 149: Although genetic factors typically define a person's sex, gender refers to how they identify on the inside. The word “gender” must be changed to “sex” throughout the manuscript. Unless authors can divide athletes according to gender identity (for example, transgender, transsexual, transvestite and etc.).

L 254: What do „total samples“ mean?

L 259, 265…..: You don't have to write statistics in the Discussion.

L 433: I suggest you can change “national competitions, etc.” to “National Competitions”.

Overall, this study is relevant and necessary. The article itself is interesting and valuable. However, specific corrections (major changes) of the manuscript must be done.

Best Regards

Reviewer 3 Report

The impact of the electronic scoring system on outcome fairness in taekwondo matches: the case of two national contexts

The aim of the study was to analyse the relationship between the color protectors and success in 1,155 international taekwondo (TKD) matches

Main Concerns:

- Like 19 and 20 stated “no relationship 19 emerged between the color protectors and the match outcome in the four editions of the SC” But yet the next line 20-21, it was stated that  “males showed positive relationships between the color blue/red and winning the match in 3 and 1 of 16 weight categories, respectively”. Aren’t these 2 sentences contradictory? And the line 22-23 also showed there is a relationship between colors and winning matches in women. Authors need to explained to clearly separate the pooled data and data from each tournament as well as data that is analysed based on gender and based on the tournament (i.e., Italian and Uzbekistan).

- authors would need to discuss in grater detail why the pooled data showed no relationship between color and winning matches; but when broken into gender (i.e. male and female) there is relationship between color and winning matches. What is that women and men (i.e., the referees/judges) differs when exposed to the color red and blue. Similarly when data was broken to Italian and Uzbekistan, there is relationship between color and winning matches – This is clearly a cultural issue or differences in culture that need to be discuss in grater detail in the Discussion section.

- and finally what is the practical implication of the present results. Maybe to adopt neutral colors during competition.

Minor issue:

- the title should include the term of color protectors. The focus should be on the fairness or non-bias of the 2 different colors during a duel contest in TKD rather than the on the ‘electronic scoring system”. The current title seems to focus on the electronic scoring system, which is clearly not the aim of the present study.  

- Line 39. In “opponents sports” does not make sense. Should be in “duel-contest sports”.

Round 2

Reviewer 2 Report

Thank you for the opportunity to review the topic “The relationship between the color of electronic protectors and the outcome in Taekwondo matches: is there fairness in National Competitions?”

First of all, I want to congratulate the Authors on their excellent research input and the big data they have been able to analyse. In my opinion, this manuscript is really valuable and interesting.

The Authors have responded to my comments and I recommend IJERPH accept the paper. Therefore, I have only one observation in order to improve the manuscript. The Results will be difficult to read for future readers due to the abundance of statistics. I therefore suggest (by the decision of the Authors) simplifying the writing of statistical characteristics in brackets (in text) and leaving only the odds ratios, 95% CI and p-value or V and p-value.

Best Regards

Reviewer 3 Report

None.

Author Response

Best regards.